# Metabolic, inflammatory and adipokine differences on overweight/obese children with and without metabolic syndrome: A cross-sectional study

Idalia Cura–Esquivel[1], Marlene Marisol Perales-Quintana[2], Liliana Torres-González[3], Katia Guzmán-Avilán[1], Linda Muñoz-Espinosa[3], Paula Cordero-Pérez[3]*

1 Departamento de Pediatría, Hospital Universitario "Dr. José E. González", Universidad Autónoma de Nuevo León, Monterrey, Nuevo León, México, 2 Departamento de Fisiología, Facultad de Medicina, Universidad Autónoma de Nuevo León, Monterrey, Nuevo León, México, 3 Unidad de Hígado, Hospital Universitario "Dr. José E. González", Universidad Autónoma de Nuevo León, Monterrey, Nuevo León, México

☯ These authors contributed equally to this work.

* paucordero@yahoo.com.mx

## Abstract

### Background

Obesity is associated with low-grade inflammation and metabolic syndrome (MetS) in both children and adults. Our aim was to describe metabolic, inflammatory and adipokine differences on overweight/obese children with and without MetS.

### Methods

This was an observational study. A total of 107 children and adolescents aged 6–18 years were included. Among this sample, n = 21 had normal body weight, n = 22 had overweight/obesity without MetS, and n = 64 had overweight/obesity with MetS. Anthropometric data and biochemical, adipokine, and inflammatory markers were measured. Different ratios were then assessed for estimate the probability of MetS. ROC analysis was used to estimate the diagnostic accuracy and optimal cutoff points for ratios.

### Results

Serum CRP levels were higher among children with overweight/obesity with MetS. Adipokines like PAI-1 and leptin were significantly lower in children with normal body weight. The Adipo/Lep ratio was highest in the group with normal body weight. TG/HDL-C and TC/HDL-C ratios were significantly correlated with BMI, DBP, PCR, and PAI-1. TC/HDL-C ratio was significantly correlated with SBP and resistin. TGL/HDL-C ratio was significantly correlated with waist and hip circumferences, fasting glucose, and MCP-1. The AUC for TG/HDL-C at the optimal cutoff of 2.39 showed 85.71% sensitivity and 71.43% specificity. CT/HDL-C at the optimal cutoff of 3.70 showed 65.08% sensitivity and 81.82% specificity. Levels of both ratios increased significantly as additional MetS criteria were fulfilled.

**Data Availability Statement:** All relevant data are within the paper and its Supporting Information files.

**Funding:** The author(s) received no specific funding for this work.

**Competing interests:** The authors have declared that no competing interests exist.

## Conclusion

Low-grade inflammation is correlated with MetS in children with overweight/obesity. TGL, HDL-C and TGL/HDL-C ratio, obtainable from routine lab tests, allows identification of MetS in children with overweight or obesity.

## Introduction

Childhood obesity is a global public health problem. Obesity, a state of chronic low-grade inflammation, results from accumulation of visceral fat, which leads to complications such as metabolic syndrome (MetS). There is currently no clear consensus on the definition of pediatric MetS [1]; however, the term refers to a set of metabolic risk factors that include obesity, dyslipidemia, hypertension, and type 2 diabetes mellitus [2]. The prevalence of MetS among children in Mexico has been reported to be as high as 54.6% [3]. Its increase in recent decades has also raised the prevalence of associated comorbidities and, since it is also considered a predictor of cardiometabolic diseases in adulthood, identification and early therapeutic intervention are crucial. It has been postulated that peripheral insulin resistance (IR) and abdominal obesity are the main factors contributing to MetS, and that its metabolic changes affect lipid metabolism due to increased low-density lipoprotein cholesterol (LDL-C), decreased high-density lipoprotein cholesterol (HDL-C), increased triglycerides (TGL), and increased fatty acids [4].

Obesity is also linked to changes in serum lipoproteins, which are in turn associated with the development of atherosclerosis. Evidence suggests that the atherosclerotic process begins in childhood. The prevalence of atherogenic dyslipidemia is increasing among children and adolescents with obesity, and is characterized by hypertriglyceridemia, increased very-low-density lipoprotein cholesterol (VLDL-C), and reduced HDL-C; its association with MetS also increases cardiovascular disease risk [5].

In adults, the relation between lipids such as TGL and HDL-C and the ratio of total cholesterol (TC) and HDL-C are widely used to assess MetS. These ratios indicate balance between all atherogenic cholesterols (including VLDL-C and HDL-C) and are thus important determinants of cardiovascular risk.

Obesity is related to both classic and novel risk factors, including prothrombotic factors (fibrinogen, plasminogen activator inhibitor-1 [PAI-1], homocysteine), inflammatory factors (interleukin 10 [IL-10], interleukin 6 [IL-6], tumor necrosis factor alpha [TNF-α], monocyte chemoattractant protein 1 (MCP-1), C-reactive protein [CRP]), and some adipocytokines (leptin, adiponectin) [6, 7]. Inflammation arising from adipose tissue has been identified as an important source of systemic inflammation and may be associated with IR. The complex MetS pathophysiology is also associated with hormone (adipokines) changes and inflammatory markers [8].

Adiponectin plays a protective role against IR and cardiovascular diseases (CVD) [9]. In contrast, leptin has proinflammatory effects; high levels are associated with development of IR and CVD. Hypoadiponectinemia and hyperleptinemia are observed in both adults and children with obesity, and the adiponectin/leptin (Adipo/Lep) ratio has been proposed as a sensitive MetS marker in children and adolescents [10].

The state of low-grade inflammation in obesity is exacerbated in individuals with MetS. Specifically, increased levels of inflammatory markers, including CRP, have been detected in children, adolescents, and adults with obesity and MetS [11]. Thus, recent attention has

focused on the relations between inflammation and hormonal dysfunction (adipokines), and their relations with MetS. As such, the objective herein was to describe metabolic, inflammatory and adipokine differences on overweight/obese children with and without MetS.

## Material and methods

### Design of the study

This was a cross-sectional analytical study of children and adolescents attending pediatric consultation at the University Hospital "Dr. José Eleuterio González" of the Autonomous University of Nuevo León in Monterrey, N.L., Mexico conducted between January 2017 and December 2019. This public hospital, with 500 beds, is the largest in Northeast Mexico with patients coming principally from the State of Nuevo Leon and surrounding states in Northern Mexico (Coahuila, Tamaulipas, and San Luis Potosi).

The institutional ethics committee approved the study (PE17-00010). A detailed letter explaining the study aims was provided to all parents or guardians and informed consent was obtained.

### Study population

Three groups were included: (I) Normal weight children: healthy children with adequate weight and height for age; (II) Obese / overweight children without metabolic syndrome; (III) Obese / overweight children without metabolic syndrome. The inclusion criteria for group (II) and (III) were: younger than age 18 years; and body mass index (BMI) ≥85th percentile according to the Centers for Disease Control and Prevention (CDC). The exclusion criteria were: congenital malformation; previous diagnosis with endocrinological, kidney, or hepatic disorder; use of any medication affecting serum lipid concentration; and refusal to participate in the study.

### Definitions

Overweight and obesity were defined according to the criteria established by the CDC. Overweight was considered a BMI between the 85th and 95th percentiles. Obesity was considered a BMI ≥95th percentile.

MetS was defined according to the de Ferranti criteria [12] and was considered present when the patient met three or more of the following criteria: (I) abdominal obesity defined as a waist circumference (WC) >75th percentile; (II) hypertension defined as a blood pressure >90th percentile; (III) TGL >100 mg/dL; (IV) HDL-C <50 mg/dL; and (V) fasting glucose >100 mg/dL.

The Adipo/Lep ratio was obtained as (Serum adiponectin levels) / (Serum leptin levels). The TC/HDL-C was calculated as (Total cholesterol) / (High-density lipoprotein cholesterol). The TGL/HDL-C was obtained as (Triglycerides) / (High-density lipoprotein cholesterol). While the RCP/HDL-C was obtained as (C-reactive protein) / (High-density lipoprotein cholesterol).

### Data collection

At the hospital visit, sex and age were recorded and anthropometrics (height, weight and waist and hip circumferences) were measured. BMI was calculated as body weight (kg)/height$^2$ (m). Blood pressure was measured using a sphygmomanometer while the child was seated.

## Biochemical and inflammatory parameters

Blood samples were taken to measure biochemical and inflammatory parameters, and adipokines. TC, HDL-C, TGL, and glucose levels were determined using an ILAB-Aries self-analyzer spectrophotometer and diagnostic kits (Instrumentation Laboratory, Bedford, MA, USA) according to the supplier's specifications. Cytokine (IL-6, TNF-α, MCP-1) concentrations were measured using a commercially available enzyme-linked immunoassays (Human IL-6 Immunoassay, Quantikine ELISA Kit; Human TNF-α Quantikine ELISA Kit; and Human CCL2/MCP-1 Immunoassay, respectively, Bio-Techne, Minneapolis, MN, USA) and are reported in pg/mL.

Inflammatory marker CRP was measured by human CRP ELISA kit (Bio-Techne, Minneapolis, MN, USA) and is reported in mg/L.

Adipokine (adiponectin, leptin), resistin, and PAI-1 levels were measured using an enzyme-linked immunoassay kit. Serum leptin level was measured by human leptin ELISA, Clinical Range kit and is reported in ng/mL (BioVendor Research and Diagnostic products, Karasek, Czech Republic). Adiponectin was measured by a Human Adiponectin/Acrp30 Duo-Set ELISA kit and is reported in mg/mL and resistin was measured by a Human Resistin Quantikine ELISA Kit (both from R&D Systems, Minneapolis, MN, USA) and are reported in ng/mL. PAI-1 was measured by a PAI1 Human ELISA Kit and is reported in ng/mL (Thermo Fisher Scientific, Waltham, MA, USA).

Ratios previously described within populations of patients who with overweight and obesity were also evaluated: Adipo/Lep, TC/HDL-C, TGL/HDL-C.

## Statistical analysis

Analyses were performed using GraphPad Prism software (v. 6.0; GraphPad, San Diego, CA, USA) or SPSS software (v.22.0; Chicago, Ill., USA) and MedCalc Statistical Software version 20.009 (MedCalc Software bvba, Ostend, Belgium). Normally distributed variables are presented as means and standard deviations and were analyzed by ANOVA-tests. Non-normally distributed variables are presented as medians and interquartile ranges and were compared by Kruskal-Wallis tests.

Bivariate and multivariate logistic regression analyses were conducted to determine factors associated with MetS, variables with $p < 0.05$ in bivariate analysis were included in multivariate analysis.

Receiver operating characteristic (ROC) analysis was performed to determine the area under the curve (AUC) to assess the precision of the TGL/HDL and TC/HDL ratios for identify children with overweight/obesity, with and without MetS. To determine the optimal cutoff point, the Younden index was used. Sensitivity and specificity of the cutoff points were calculated. A correlation study for TGL/HDL-C and TC/HDL-C was carried out using the Spearman correlation. For all analyses, $p < 0.05$ was considered statistically significant.

## Results

### Sample characteristics

The total sample was 107 patients, among whom 63 were male (58.80%) and 44 were female (41.10%); their mean age was 10.52(1.76) years. Among the total sample, 21 children (19.60%) had normal body weight and 86 (80.40%) had overweight/obesity.

### MetS diagnosis

Among the study sample, 64 (59.81%) had obesity or overweight and met the MetS diagnostic criteria of three of the five Ferranti criteria. In this subgroup, 76.64% had abdominal obesity,

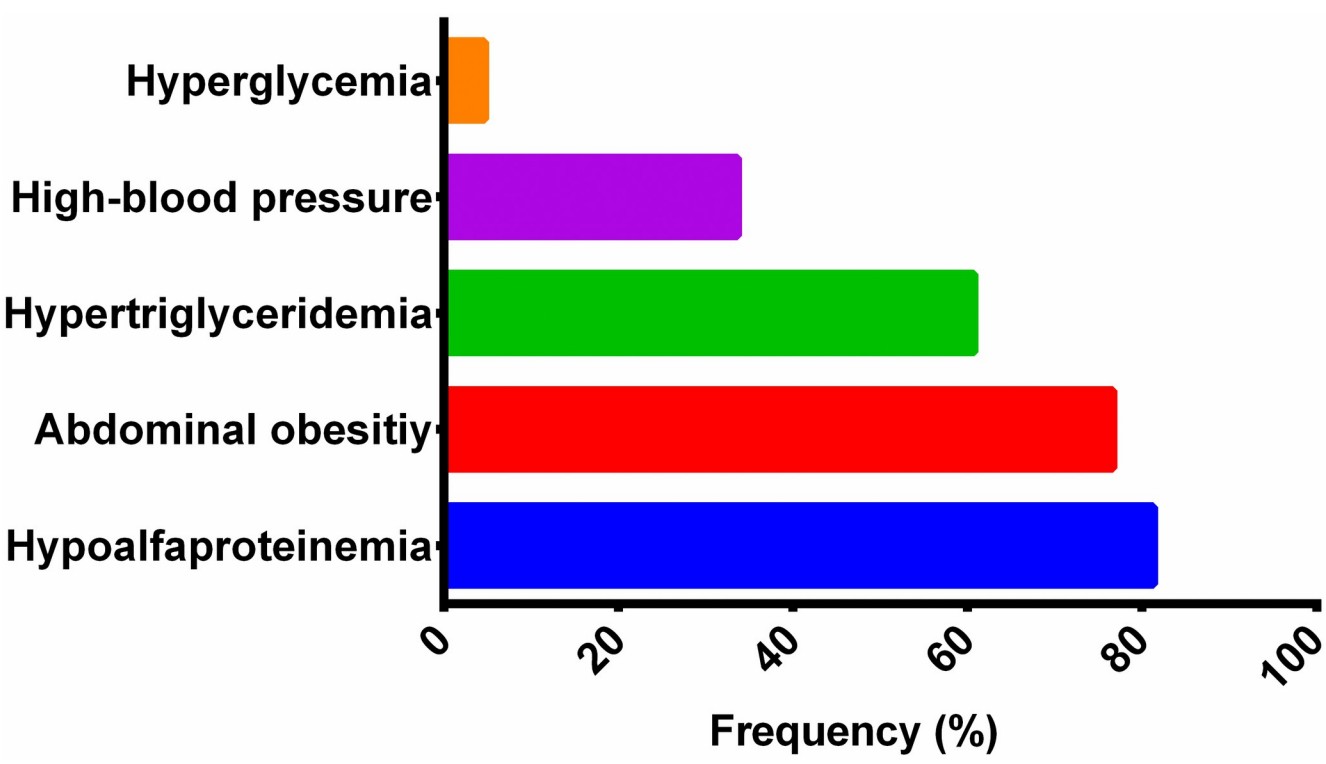

**Fig 1. Frequency of the Ferranti criteria used for the diagnosis of metabolic syndrome.**

33.64% presented arterial hypertension, 60.75% had elevated TGL levels, 81.31% had low HDL-C levels, and 4.67% presented with hyperglycemia (Fig 1).

### Anthropometric, biochemical, adipokine, and cytokine characteristics

Sample anthropometric, biochemical, adipokine, and cytokine characteristics are described in Table 1. Compared with children with normal body weight, those with overweight/obesity and MetS had significantly higher BMI, systolic blood pressure (SBP), diastolic blood pressure (DBP), TGL, and MCP-1, and significantly lower HDL-C and TGL/HDL-C ratio (4.51(3.15); p<0.0001). There were no significant differences on indices between the children with overweight/obesity without MetS and those with normal body weight (p>0.05) (Table 1).

After bivariate analysis, we found that children with high levels of HDL-C have lower probability of having MetS (OR = 0.88, 95% CI = 0.81–0.95, p = 0.002). In addition, children with high levels of TGL have lower probability of having MetS (OR = 1.022, 95% CI = 1.01–1.033, p<0.001).

Serum leptin, resistin, PAI-1, and CRP levels differed significantly between children with normal body weight and those with overweight/obesity, and between those with and without MetS (all p<0.05), being higher in children with overweight/obesity and MetS.

The highest adiponectin levels were in children with normal body weight (31.13; IQR 12.46–36.90 mg/mL) and the lowest were in children with overweight/obesity with MetS (23.82; IQR 15.56–32.87 mg/mL). In contrast, the highest leptin levels were in children with overweight/obesity without MetS (18.17; IQR 11.74–29.17 ng/mL) and the lowest were in those with normal body weight (2.31; IQR 1.47–4.90 ng/mL) (Table 1). Consequently, the Adipo/Lep ratio was highest in children with normal body weight (6.68(7.89)), lower in those

**Table 1. Anthropometric and laboratory parameters of the children.**

| | Normal weight children (n = 21) | Obese/overweight children without metabolic syndrome (n = 22) | Obese/overweight children with metabolic syndrome (n = 64) | p |
|---|---|---|---|---|
| Age in years, mean (age range) | 8.70 (6–11) | 10.00 (6–15) | 11.00 (8–15) | |
| Sex Male/Female, n (%) | 12/9 (57.14/42.86) | 14/8 (63.64/36.36) | 37/27 (57.81 / 42.19) | |
| Obese / Overweight | | 17/5 (77.27 / 22.73) | 54/10 (84.38 / 15.63) | |
| **Anthropometric variables, mean (SD)** | | | | |
| Height (cm) | 134.00 (9.30) | 149.00 (11.00) | 147.00 (8.70) | < 0.001 † ‡ |
| Weight (kg) | 28.00 (4.80) | 61.00 (17.00) | 64.00 (16.00) | < 0.001 † ‡ |
| BMI (kg/m$^2$) | 16.00 (2.10) | 27.00 (4.50) | 29.00 (5.30) | < 0.001 † ‡ |
| Waist circunference (cm) | 57.00 (6.40) | 91.00 (12.00) | 89.00 (9.20) | < 0.001 † ‡ |
| Hip circunference(cm) | 67.00 (7.40) | 97.00 (12.00) | 99.00 (11.00) | < 0.001 † ‡ |
| **Blood pressure, mean (SD)** | | | | |
| SBP (mmHg) | 98.00 (7.90) | 108.00 (11.00) | 111.00 (16.00) | 0.001 † ‡ |
| DBP (mmHg) | 60.00 (8.00) | 59.00 (10.00) | 69.00 (10.00) | < 0.001 ‡ § |
| **Biochemical variables, median (IQR)** | | | | |
| Total cholesterol (mg/dL) | 150.0 (130.0–173.0) | 141.00 (116.0–160.0) | 157.00 (134.0–184.0) | 0.075 |
| HDL cholesterol (mg/dL) | 49.00 (55.00–41.00) | 46.00 (34.00–51.00) | 39.00 (34.00–43.00) | < 0.001 ‡ § |
| Triglycerides (mg/dL) | 76.00 (57.00–99.00) | 75.00 (51.00–101.00) | 183.00 (123.00–223.00) | < 0.001 ‡ § |
| Fasting glucose (mg/dL) | 80.00 (78.00–85.00) | 78.00 (74.00–85.00) | 83.00 (77.00–87.00) | 0.052 |
| CRP (mg/L) | 0.10 (0.10–0.38) | 0.50 (0.15–0.89) | 1.10 (0.33–3.50) | < 0.001 † ‡ |
| **Adipokines, median (IQR)** | | | | |
| Adiponectin (mg/mL) | 31.13 (12.46–36.90) | 25.42 (17.90–36.83) | 23.82 (15.56–32.87) | 0.258 |
| Resistin (ng/mL) | 17.43 (12.05–19.60) | 19.68 (16.55–29.30) | 21.35 (18.31–28.61) | 0.020 ‡ |
| PAI-1 (ng/mL) | 19.18 (15.31–22.58) | 27.61 (23.20–37.38) | 27.37 (17.89–41.26) | <0.001 † ‡ |
| Leptin (ng/mL) | 2.31 (1.47–4.90) | 18.17 (11.74–29.17) | 16.89 (2.52–23.62) | < 0.001 † ‡ |
| **Cytokines, median (IQR)** | | | | |
| IL-6 (pg/mL) | 16.76 (16.66–17.16) | 17.16 (16.71–18.01) | 17.60 (16.76–18.75) | 0.007 ‡ |
| TNF-α (pg/mL) | 17.65 (13.51–20.31) | 17.65 (16.02–21.45) | 18.68 (16.30–127.30) | 0.018 ‡ |
| MCP-1 (pg/mL) | 331.6 (182.7–405.3) | 279.8 (395.4–231.3) | 372.9 (272.2–770.60) | 0.033 ‡ § |
| **Ratio, mean (SD)** | | | | |
| Adipo/Lep ratio | 6.68 (7.89) | 1.08 (1.62) | 1.08 (0.85) | < 0.001 † ‡ |
| RCP/HDL-C ratio | 0.002 (0.001) | 0.01 (0.01) | 0.02 (0.05) | < 0.001 † ‡ |
| CT/HDL-C ratio | 3.08 (0.98) | 3.28 (0.53) | 4.24 (1.16) | < 0.001 ‡ § |

(*Continued*)

**Table 1.** (Continued)

| | Normal weight children (n = 21) | Obese/overweight children without metabolic syndrome (n = 22) | Obese/overweight children with metabolic syndrome (n = 64) | p |
|---|---|---|---|---|
| TGL/HDL-C ratio | 1.46 (0.91) | 1.78 (1.84) | 4.51 (3.15) | < 0.001 ‡ § |

‡ showed significant difference between obese/overweight with metabolic syndrome and normal weight children; † showed significant difference between obese/overweight without metabolic syndrome and normal weight children

§ showed significant difference between obese/overweight with metabolic syndrome and obese/overweight without metabolic syndrome.

with overweight/obesity, and not significantly different between those without and with MetS (1.08(1.62) vs 1.08(0.85), respectively) (Table 1).

Serum CRP levels in children with normal body weight (0.10; IQR 0.10–0.38 mg/L) were significantly lower than in those with obesity and MetS (1.10; IQR 0.33–3.50 mg/L, p<0.0001).

Among the ratios evaluated (Table 1), only TC/HDL-C and TGL/HDL-C differed significantly between children with overweight/obesity with and without MetS (p<0.0001).

TC/HDL-C and TGL/HDL-C ROC curve analyses differentiated between children with overweight/obesity with MetS, as shown in Fig 2 and Table 2. TGL/HDL-C had the highest probability for MetS with an AUC of 0.85 and a cutoff value >2.39. Correlations between TG/HDL-C and TC/HDL-C and important variables are shown in Table 3. Both ratios were correlated with BMI, DBP, PCR, and PAI-1. Only the TC/HDL-C ratio was significantly correlated with SBP and resistin. Only the TGL/HDL-C ratio was significantly correlated with waist and hip circumferences, fasting glucose, and MCP-1.

## Discussion

Obesity-related diseases and complications were previously considered exclusive to the adult population, so evaluating them in childhood or adolescence was not routine practice. Nevertheless, many studies have now shown that childhood obesity tends to perpetuate into adulthood; this favors early development of metabolic disease and increases risk for CVD and diabetes, which in turn decreases life expectancy [13, 14].

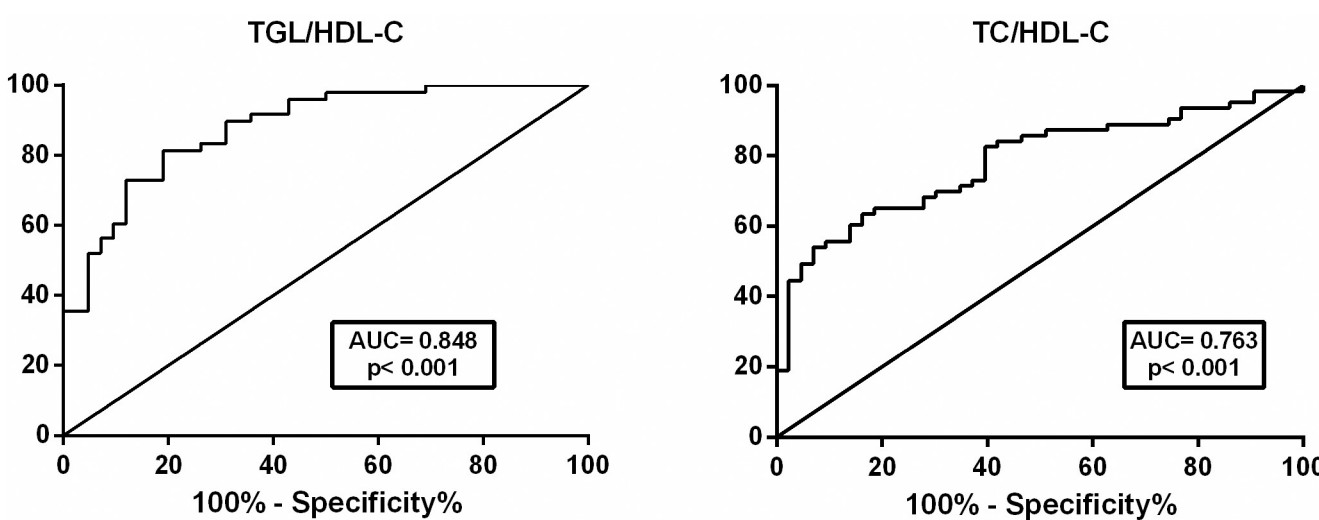

**Fig 2. ROC curves for TGL/HDL-C and TC/HDL-C ratios in predicting metabolic syndrome in children with overweigh or obesity.**

**Table 2. AUCROC and cutoff values of TGL/HDL-C and TC/HDL-C ratios to estimate probability of metabolic syndrome in obese and overweight children.**

| | Area under the ROC curve | | | | Younden index | | | |
|---|---|---|---|---|---|---|---|---|
| | Area | Error | 95% CI | p | Criterion | J | Sensivity | Specificity |
| TC/HDL-C ratio | 0.76 | 0.05 | 0.66–0.85 | <0.001 | >3.71 | 0.47 | 65.08 | 81.82 |
| TGL/HDL-C ratio | 0.85 | 0.05 | 0.75–0.92 | <0.001 | >2.39 | 0.57 | 85.71 | 71.43 |

Herein, we evaluated children with overweight/obesity and identified that in addition to abnormal weight, they typically present with at least one metabolic alteration, of which the most prevalent are dyslipidemia (60.75%) and arterial hypertension (33.64%).

The prevalence of pediatric MetS is variable and depends on the diagnostic criteria and component cutoff values [1]. We found a MetS prevalence of 58.81% using the Ferranti criteria, consistent with the meta-analysis by Bitew et al. showing a general prevalence of 56.32% in studies using the same criteria [15]. The most prevalent MetS criterion herein were low HDL-C (81.31%), central obesity (76.64%), and high total TGL (60.75%).

Obesity is a state of chronic low-grade inflammation, in which nutrient overload, increased metabolic demands, and lipotoxicity at the adipose level contribute to production of inflammatory mediators. Some of these markers are synthesized by adipocytes, including acute phase proteins such as CRP, haptoglobin, PAI-1, TNF-α, resistin, and cytokines (IL-1b, IL-6, IL-8, IL-10) [16, 17]. MetS is characterized by multiple cardiovascular risk factors; the endothelial dysfunction from this prothrombotic, inflammatory state is caused by the expression of inflammatory cytokines and cell adhesion molecules [17, 18].

As the main inhibitor of fibrinolysis, high levels of PAI-1 can increase coronary heart disease risk. Increased PAI-1 is involved with control of insulin signaling in adipocytes and can be considered a component of MetS [19]. IR states have been associated with elevated PAI-1 levels and altered plasma lipids, which helps explain the characteristic prothrombotic state of

**Table 3. Correlation between TGL/HDL-C and TC/HDL-C ratios with variables.**

| | TC/HDL-C ratio | | | TGL/HDL-C ratio | | |
|---|---|---|---|---|---|---|
| | r | p | CI– 95% | r | p | CI-95% |
| **Pearson** | | | | | | |
| Waist circumference (cm) | 0.192 | 0.054 | -0.003 to 0.374 | 0.353 | <0.001 | 0.169 to 0.514 |
| Hip circumference(cm) | 0.153 | 0.125 | -0.043 to 0.339 | 0.410 | < 0.001 | 0.232 to 0.561 |
| BMI (kg/m²) | 0.264 | 0.006 | 0.077 to 0.433 | 0.440 | < 0.001 | 0.169 to 0.514 |
| SBP (mmHg) | 0.304 | 0.003 | 0.109 to 0.476 | -0.056 | 0.589 | -0.256 to 0.148 |
| DBP (mmHg) | 0.356 | <0.001 | 0.166 to 0.520 | 0.274 | 0.008 | 0.075 to 0.451 |
| **Spearman** | | | | | | |
| Fasting glucose (mg/dL) | 0.167 | 0.086 | -0.030 to 0.352 | 0.247 | 0.011 | 0.052 to 0.424 |
| PCR (mg/L) | 0.207 | 0.039 | 0.005 to 0.392 | 0.611 | < 0.001 | 0.466 to 0.724 |
| Adiponectin (mg/mL) | -0.115 | 0.244 | -0.305 to 0.084 | -0.149 | 0.130 | -0.338 to 0.0502 |
| Resistin (ng/mL) | 0.203 | 0.037 | 0.006 to 0.385 | 0.112 | 0.258 | -0.088 to 0.303 |
| PAI-1 (ng/mL) | 0.236 | 0.015 | 0.040 to 0.414 | 0.232 | 0.018 | 0.036 to 0.412 |
| Leptin (ng/mL) | 0.077 | 0.438 | -0.123 to 0.271 | 0.055 | 0.580 | -0.146 to 0.251 |
| IL-6 (pg/mL) | 0.034 | 0.732 | -0.164 to 0.228 | 0.127 | 0.197 | -0.072 to 0.316 |
| TNF-α (pg/mL) | 0.078 | 0.456 | -0.133 to 0.283 | -0.062 | 0.559 | -0.269 to 0.151 |
| MCP-1 (pg/mL) | 0.069 | 0.487 | -0.132 to 0.265 | 0.199 | 0.045 | -0.0008 to 0.384 |

Levels of both ratios increased significantly as increasing numbers of Ferranti criteria were fulfilled as shown in Fig 3.

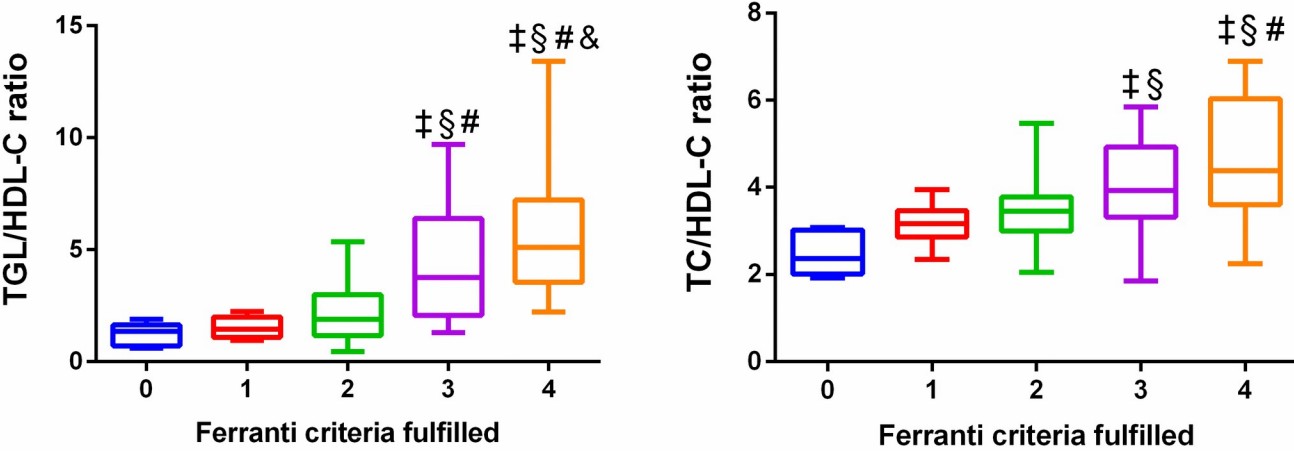

**Fig 3. Levels of TGL/HDL-C and TC/HD-C ratios in relation to the number of fulfilled Ferranti criteria.** ‡ showed significant difference vs 0 Ferranti criteria fulfilled; § showed significant difference vs 1 Ferranti criteria fulfilled; # showed significant difference vs 2 Ferranti criteria fulfilled; & showed significant difference vs 0 Ferranti criteria fulfilled.

these pathologies [19, 20]. Children in our sample with overweight or obesity showed elevated PAI-1 levels compared with those with normal body weight, and similarly elevated TGL and blood pressure. In contrast, PAI-1 levels did not differ between those with and without MetS, providing further evidence of its association with plasma lipids but not necessarily MetS.

However, MCP-1 levels were considerably elevated in children with obesity and MetS compared with those with normal body weight. MCP-1 levels were also correlated with waist and hip circumferences, BMI, and DBP. Kim et al. found similar correlations with BMI and WC in young Koreans [21].

Resistin, a protein suspected to be related to obesity and IR, is reportedly increased in children with central obesity [22]. Herein, resistin was elevated in children with overweight/obesity, both with and without MetS; however, the only significant difference was between children with MetS and those with normal body weight. That no difference was found between the groups with obese/overweight with and without MetS suggests that plasma resistin may be a weak biochemical marker of metabolic dysfunction. This supports the notion that only a small proportion of variance in resistin can be explained by MetS-related factors.

Although central obesity assessed with WC is considered a better marker of metabolic risk than high BMI in adults, pediatric results have been contradictory [23, 24]. Our subsample with overweight/obesity showed significant differences on various inflammatory markers, and those with abdominal obesity had higher CRP levels compared with those without. Findings were consistent for PAI-1 and resistin, but not MCP-1, suggesting that WC may be correlated with inflammation and metabolic risk regardless of MetS status.

Other adipokines evaluated herein were adiponectin and leptin. While leptin acts primarily in the hypothalamus to control food intake, satiety, and energy expenditure, adiponectin is associated with reduced total body fat mass and promotes insulin sensitivity [25, 26]. Obesity and MetS are characterized by decreased serum adiponectin in parallel with increased concentrations of circulating leptin. Consequently, the Adipo/Lep ratio is associated with BMI and MetS status [26, 27]. The results herein show a negative correlation between Adipo/Lep ratio and BMI, meaning that Adipo/Lep ratio is significantly lower in children and adolescents with obesity, with or without MetS, compared with children with normal body weight. This biomarker decreases with increasing metabolic risk factors, which is why it has been proposed as predictive of MetS [25, 26].

Herein, adiponectin concentrations were lower in children with overweight/obesity with MetS compared with those without MetS, providing further evidence that adiponectin decreases in the presence of previously identified MetS parameters [28].

Dyslipidemia, particularly TC and TGL levels, has been described as an important risk factor for CVD, based on various indices [29, 30]. TC and TGL reflect the concentrations of the lipoproteins that transport them. HDL-C has antiatherogenic activity. Together, TC/HDL-C and TGL/HDL-C ratios may reflect the balance between these lipoproteins and could serve as a useful marker for cardiovascular risk. These relations have been evaluated in adults and children with and without obesity [31–33].

In an otherwise adult healthy Mexican sample, TGL/HDL-C ratio was associated with low insulin sensitivity and MetS, suggesting that it may serve as a reference index for MetS [32]. Herein, we found that TGL/HDL-C ratio was higher in children with overweight/obesity compared with children without overweight/obesity. When evaluating this in children with obesity with and without MetS, the association was stronger in the presence of MetS. This index also rose with increased numbers of fulfilled MetS criteria. In bivariate analysis and multivariate logistic regression, only HDL-C and TGL showed a significant correlation with MetS. This confirms that both biochemical markers are relevant in the pathophysiology of this syndrome, thus contributing to the usefulness of TC/HDL-C and TGL/HDL-C ratios for MetS screening in obese children, such as has previously been reported [34].

The TGL/HDL-C ratio was predictive of MetS with an AUC of 0.848 (95% confidence interval [CI]: 0.753–0.917). The optimal cutoff value was >2.390. Our TGL/HDL-C cutoff value for MetS identification was higher than the values of 1.25 reported for Chinese children with obesity [34] and 2.0 for Korean children with overweight [35]. These differences may be attributable to population-based ethnic and genetic variations. Herein, TGL/HDL-C ratio was correlated with BMI, WC, fasting glucose, and the inflammatory parameters CRP and PAI-1, suggesting its value for identifying MetS.

Elevated TC/HDL-C has been associated with a proinflammatory state in adults and adolescents, as it is strongly related to elevated CRP levels [33]. Herein, TC/HDL-C was evaluated as an indicator of cardiovascular risk. Consistent with other studies, it was significantly correlated with BMI, hypertension, and systemic inflammatory parameters: CRP, resistin, and PAI-1. This evidence confirms TC/HDL-C as a significant low-grade inflammation parameter. As such, it can be used to predict cardiovascular risk, as it reflects an imbalance between cholesterol transported by atherogenic lipoproteins and protective lipoproteins. It is widely accepted that obesity induces lipid biochemistry alterations in the development of atherogenic dyslipidemia, a critical factor in cardiovascular events among adults. The presence of atherosclerotic plaques has been reported in autopsies of children as young as age two years in the Bogalusa Heart Study [36]. Early identification is a crucial step toward reducing related morbidity and mortality. Herein, metabolic risk factors like obesity, atherogenic dyslipidemia (high TGL, low HDL-C), and high blood pressure were the most common MetS parameters (Fig 1).

RCP levels differed significantly between children with and without MetS, and were positively correlated with metabolic risk factors such as TGL/HDL-C and TC/HDL-C. It is known that childhood dyslipidemia can trigger low-grade inflammation even in the absence of obesity, since these parameters are increased even in children who without overweight. However, in children with obesity, these parameters are considerably increased in the presence of MetS.

In sum, the main risk factors correlated with metabolic disease and cardiovascular risk were WC, hypertension, atherogenic dyslipidemia (elevated TC, low HDL-C), HDL-C, TGL and inflammatory parameters (CRP, PAI-1). Of note, HDL-C, TGL, TGL/HDL-C ratio and TC/HDL-C ratio are available from routine lab tests, simplifying surveillance for cardiovascular risk among children with overweight or obesity.

To date, few studies in Mexico have evaluated the cardiovascular risk indices in children. Therefore, some limitations of the present study must be acknowledged. This study was based on a population of children and adolescents who attended an obesity consultation motivated by themselves or their parents to receive treatment, consequently, the results cannot be generalized since the sample consisted mainly of children of a medium-low socioeconomic level who were more predisposed to the development of metabolic disorders. Moreover, a weakness of this study was the comparison of our group of children with populations of different ethnic groups, making it difficult to compare the results obtained with those of other authors, especially considering the population differences and various methodologies used. Despite these limitations, we must highlight that Mexico is one of the first places in childhood obesity; therefore, this study is extremely important for the recognition of risk factors since childhood that influence the appearance of chronic-degenerative diseases in adulthood.

In conclusion in the population evaluated HDL-C, TGL, TGL/HDL-C ratio and TC/HDL-C ratio shown major alteration in overweight and obese children with MetS, which can be explained because the lipid parameters are part of the MetS diagnostic criteria; however, the inflammatory parameters and adipokines evaluated did not shown a difference in overweight or obese children with/without MetS, confirming the chronic inflammation state that has been previously described in patients under these conditions.

## Supporting information

**S1 Checklist. STROBE statement—checklist of items that should be included in reports of observational studies.**
(DOCX)

**S1 File.**
(XLSX)

**S2 File.**
(SPV)

## Acknowledgments

We acknowledge all staff and patients who offered help for this study.

## Author Contributions

**Conceptualization:** Idalia Cura–Esquivel, Marlene Marisol Perales-Quintana.

**Formal analysis:** Idalia Cura–Esquivel, Marlene Marisol Perales-Quintana.

**Investigation:** Idalia Cura–Esquivel, Marlene Marisol Perales-Quintana.

**Methodology:** Idalia Cura–Esquivel, Marlene Marisol Perales-Quintana, Katia Guzmán-Avilán.

**Project administration:** Idalia Cura–Esquivel, Marlene Marisol Perales-Quintana.

**Resources:** Idalia Cura–Esquivel, Marlene Marisol Perales-Quintana, Liliana Torres-González, Paula Cordero-Pérez.

**Supervision:** Idalia Cura–Esquivel, Marlene Marisol Perales-Quintana, Linda Muñoz-Espinosa, Paula Cordero-Pérez.

**Validation:** Idalia Cura–Esquivel, Marlene Marisol Perales-Quintana, Paula Cordero-Pérez.

**Writing – original draft:** Idalia Cura–Esquivel, Marlene Marisol Perales-Quintana, Liliana Torres-González, Katia Guzmán-Avilán, Linda Muñoz-Espinosa, Paula Cordero-Pérez.

**Writing – review & editing:** Idalia Cura–Esquivel, Liliana Torres-González, Katia Guzmán-Avilán, Linda Muñoz-Espinosa.

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
