## [Decision Letter · Decision Letter 0]

2 Aug 2022

PONE-D-22-12285Assessment of cardiometabolic risk in children with obesity: adipokines, lipoprotein ratios, and inflammatory markersPLOS ONE

Dear Dr. Cordero Pérez,

Thank you for submitting your manuscript to PLOS ONE. After careful consideration, we feel that it has merit but does not fully meet PLOS ONE’s publication criteria as it currently stands. Therefore, we invite you to submit a revised version of the manuscript that addresses the points raised during the review process.

We look forward to receiving your revised manuscript.

Kind regards,

Omar Yaxmehen Bello-Chavolla, MD, PhD

Academic Editor

PLOS ONE

Journal Requirements:

Could you therefore please include the title page into the beginning of your manuscript file itself, listing all authors and affiliations

3. Please ensure that you include a title page within your main document. You should list all authors and all affiliations as per our author instructions and clearly indicate the corresponding author.

Additional Editor Comments (if provided):

Both reviewers suggested significant edits prior to reconsideration of the manuscript, I believe a major revision would address most of these issues.

Reviewers' comments:

Reviewer's Responses to Questions

**Comments to the Author**

1. Is the manuscript technically sound, and do the data support the conclusions?

Reviewer #1: Partly

Reviewer #2: Partly

2. Has the statistical analysis been performed appropriately and rigorously? 

Reviewer #1: No

Reviewer #2: No

3. Have the authors made all data underlying the findings in their manuscript fully available?

Reviewer #1: No

Reviewer #2: No

4. Is the manuscript presented in an intelligible fashion and written in standard English?

Reviewer #1: Yes

Reviewer #2: Yes

5. Review Comments to the Author

Reviewer #1: The present work entitled "Assessment of cardiometabolic risk in children with obesity: adipokines, lipoprotein ratios, and inflammatory markers", which aimed "to explore the profile of adipokines, inflammatory markers, and lipid ratios to identify children with cardiometabolic risk" shows an interesting knowledge gap that could be useful in the search for information on cardiovascular risk, however, the submitted manuscript has various errors that must be resolved during a review.

My comments are as follows

Major Comments:

1.- The objective of the study is very ambiguous, "to explore the profile of adipokines, lipoprotein ratios, and inflammatory markers", with cardiometabolic risk, it is interpreted that each of these variables is associated with "cardiometabolic risk", but never cardiometabolic risk was defined, but only mention is made of “cardiometabolic risk factors”, so the objective and the rest of your work must be adjusted to a specific objective of what is sought to be carried out, whether it is the case of associating certain variables independent with cardiometabolic risk factors or with metabolic syndrome, as is.

2.- The type of study is not very clear, it is only mentioned as an observational study, but it really seems that it is a cross-sectional analytical study, so it should be classified that way, or give reasons why it is another type of study.

3.- During the manuscript there is talk about cardiometabolic risk, but in reality only metabolic syndrome was explored, the introduction and discussion should be changed so that the terms and references are appropriate to what was done, following the definitions of the study itself.

4.- It must be defined how each of the presented indices and the units of the variables used for their calculation were calculated, this is because these are their independent variables, and they should be considered with importance throughout the manuscript, and identified as such, so that the reader can understand what the authors want to do.

In the statistical analyzes it is mentioned that only comparisons were applied through the Student's t and Mann-Whitney U tests, but in the tables 3 groups are being presented in which the sample was divided, the comparisons should not be made only by bivariate tests, but by analysis of variance or the Kruskall-Wallis test if the ANOVA assumptions are not met.

5.- Table 3 mentions the word "prediction" referring to a prediction of metabolic syndrome by lipoprotein indices, but in reality they do not make a prediction, only an estimate of the probability of presenting metabolic syndrome at the same time of the measurement of the variables, and a prediction is related to the probability of estimating the presentation of a future condition. Therefore, I recommend avoiding the term throughout the manuscript.

6.- The reason why the correlations of the lipoprotein indices with all the variables were made is not understood, it would be interesting if the authors explained them and if it was part of a secondary objective, mention it as such in the text.

Minor Comments:

1.- The title of the study is ambiguous, because once the body of the manuscript is read, it is observed that the population studied is not only obese children but also overweight and obese, in the same way a cardiometabolic risk assessment was not made per se, but only its association with the metabolic syndrome. The function for which the words “adipokines, lipoprotein ratios, and inflammatory markers” are placed in the title is not clear, because the reader understands that these markers are the ones associated with cardiometabolic risk, but in reality it was never carried out, only the lipoprotein indices were associated with metabolic syndrome, the title will have to be restructured, following the recommendations of STROBE, as well as indicating the study population, the dependent and independent variables and the type of study.

2.- There are no references to the definitions presented, the references to each definition must be attached.

3.- Throughout the manuscript, the terms “abdominal adiposity” and “hypertension” are presented, which are mentioned in the results, but were never defined as such in the methodology, this error must be resolved.

4.- The symbol ± is used throughout the text to show the standard deviation, please avoid it and place the standard deviation only in parentheses or with the abbreviation "SD".

5.- In line 202 of the text the word associated is mentioned to show the relationship between certain variables, I consider the term “correlated” to be more optimal.

Reviewer #2: I want to thank the opportunity to review this manuscript. Cura–Esquivel Idalia and Perales-Quintana Marlene Marisol et al performed a cross-sectional study design to evaluate the metabolic, adipokine and inflammatory profile among overweight and obese pediatrics with cardiometabolic risk (classified with Mets according to the Ferranti criteria). The authors found that CRP, PAI-, Adipo-Lep and Tg/HDL and TC-HDL-C were different among patients living with MetS. Furthermore, the authors estimate the optimal threshold for Tg/HDL and TC-HDL-C to identify MetS. The novelty of this manuscript relies on the population and adipokines and inflammatory markers measured. A potential limitation is the relatively small sample size and the single center recruitment facility. Overall, the manuscript has an important message and delivers relevant findings in the field of endocrinology. Nevertheless, some findings have a lack of rational to demonstrate the objective. Furthermore, there are some methological and statistical issues that needs to be solved prior to the recommendation of acceptance of this manuscript. My suggestions are appended below.

Introduction

• A main concern of the objective of this study is that the authors shough to assess cardiometabolic risk using a proxy metabolic syndrome among children. Although it is a simplified and relevant concept, I would suggest to clearly specify in the objective that the authors used metabolic syndrome as their main dependent variable.

Methods

• Please specify the type of study the authors used (E.g., Cross-sectional recruitment).

• Please provide a more detailed context of the center of reference where the authors recruited the patients.

• For this reviewer, it is quite unclear why the authors sought to explore a cut-off point to determine TGL/HDL and TC/HDL with and without MetS. Overall, I could interpret this to explore a tress-hold to define as a proxy of insulin resistance using MetS as their outcome variable in children. Please explain the purpose of this analysis. Furthermore, I would suggest to perform a stratification of this threshold among children and adolescence.

• The authors state that part of the inclusion criteria where patients ≥85th percentile of BMI (definition of overweight), but in table 1 they displayed 21 patients with normal weight (patients <85th percentile?). Please clarify.

Results

• A main issue with table 1, is that the authors sought to compare the cardiometabolic risk using as a proxy the MetS construct, which sought to identify the profile among the subjects that fulfill the Ferranti criteria. A better approach to describe the results could be to display the overall population (n=107) comparing between subjects with MetS (n=64) and without MetS (n=43).

• Another mayor issue is that the manuscript does not fully demonstrate the objective posed by the authors. As for what this reviewer understood, the sought to describe the metabolic and biochemical characteristics of patients with either obesity/overweigh with MetS and without MetS. Then, they sought to evaluate the adipokine and inflammatory profile, for which they found that patients with MetS had increased inflammation markers and lowest Adipo/Lep, which goes in line that MetS is mainly attributable to IR. Then, to prove this, the authors sought to evaluate the use of TGL/HDL and TC/HD-C ratios, for which found a cut-off value to determinate MetS. The structure of the results seems more like the objective was to describe metabolic, inflammatory and adipokine differences on overweight/obese children with and without MetS.

• I would suggest exploring a logistic regression model to assess whether metabolic, inflammatory and adipokine parameters predicts MetS.

• For what it is observed, the higher MetS criteria was hypoalfaproteinemia and abdominal obesity. I would suggest performing a sensitivity analysis to evaluate whether the predictors identified by logistic regression, are really driven by these two main components.

• Please describe the rationale and purpose of including the variables in Table 4.

Discussion

• Overall, a well-written discussion regarding the main findings.

• A consideration would be to evaluate whether the causality of having MetS are all the metabolic dysfunctions caused by environmental or nutritional factors linked by the burden of obesity observed in Mexico.

Conclusion

• Please let the conclusion statement at the end of the manuscript.

Minor comments

• Please round all the result for two decimals

• The p-value in table 1 are not displayed in the pdf proof.

• Please include headings in the methods and result section to separate the main ideas across the manuscript.

• Suggest including the waist-to-height ratio, waist-to-hip ratio, and non-HDL cholesterol estimation as part of table 1.

• The ratios estimated in table 2 could be included as part of table 1.

• In figure 1, please modify hypertension to high-blood pressure, TGL to hypertriglyceridemia, fasting glucose to hyperglycemia, HDL to hypoalfaproteinemia. Also suggest including an additional label with the actual prevalence.

• In figure 3, please include the trending p-value to observe a potential dose-relationship progression with this two markers and Ferranti criteria.

• Please check for typos of TC-HDL-C in table 3 and 4.

6. PLOS authors have the option to publish the peer review history of their article (what does this mean?). If published, this will include your full peer review and any attached files.

Reviewer #1: **Yes: **Ashuin Kammar-García

Reviewer #2: **Yes: **Neftali Eduardo Antonio-Villa, MD

---

## [Author Response · Author response to Decision Letter 0]

19 Sep 2022

Reviewer 1

The objective of the study is very ambiguous, "to explore the profile of adipokines, lipoprotein ratios, and inflammatory markers", with cardiometabolic risk, it is interpreted that each of these variables is associated with "cardiometabolic risk", but never cardiometabolic risk was defined, but only mention is made of “cardiometabolic risk factors”, so the objective and the rest of your work must be adjusted to a specific objective of what is sought to be carried out, whether it is the case of associating certain variables independent with cardiometabolic risk factors or with metabolic syndrome, as is.

After discussing the point with the co-authors and reviewing the bibliography, we have decided to change the objective of the work since, as you have rightly commented, we evaluate risk in overweight/obese children with and without MetS. 

These changes are observed in the abstract (lines 3-5) and in the introduction (lines 74-75).

The aim of the study was changed to: Describe the metabolic, inflammatory, and adipokine differences in overweight/obese children with and without metabolic syndrome.

However, it should be noted that "cardiometabolic risk" is a term that is used to describe the metabolic alterations, mainly in lipids, that overweight/obese children have and their relationship with the appearance of early atherosclerosis in adulthood.

The type of study is not very clear, it is only mentioned as an observational study, but it really seems that it is a cross-sectional analytical study, so it should be classified that way, or give reasons why it is another type of study.

We appreciate the observation, and this has already been corrected in the material and methods section (line 78)

During the manuscript there is talk about cardiometabolic risk, but in reality only metabolic syndrome was explored, the introduction and discussion should be changed so that the terms and references are appropriate to what was done, following the definitions of the study itself.

The word “cardiometabolic risk” was replace by “metabolic risk”.

However, it should be noted that "cardiometabolic risk" is a term that is used to describe the metabolic alterations, mainly in lipids,associated with hypertension that overweight /obese children have and their relationship with the appearance of early atherosclerosis in adulthood.

It must be defined how each of the presented indices and the units of the variables used for their calculation were calculated, this is because these are their independent variables, and they should be considered with importance throughout the manuscript, and identified as such, so that the reader can understand what the authors want to do.

We appreciate the observation; we have decided to include the way to obtain these ratios (lines 107 - 111). Throughout the manuscript the word "ratio" has been added whenever reference is made to the indices.

In the statistical analyzes it is mentioned that only comparisons were applied through the Student's t and Mann-Whitney U tests, but in the tables 3 groups are being presented in which the sample was divided, the comparisons should not be made only by bivariate tests, but by analysis of variance or the Kruskall-Wallis test if the ANOVA assumptions are not met.

This analysis had already been done but there was an omission in the statistical analysis section. This has already been corrected in this material and methods section (lines 145 and 147)

Table 3 mentions the word "prediction" referring to a prediction of metabolic syndrome by lipoprotein indices, but in reality they do not make a prediction, only an estimate of the probability of presenting metabolic syndrome at the same time of the measurement of the variables, and a prediction is related to the probability of estimating the presentation of a future condition. Therefore, I recommend avoiding the term throughout the manuscript.

We appreciate the observation, we agree and have already made the change throughout the document and in the header of Table 3.

The reason why the correlations of the lipoprotein indices with all the variables were made is not understood, it would be interesting if the authors explained them and if it was part of a secondary objective, mention it as such in the text.

This analysis was used with the objective of showing that these indices are indeed useful to suspect or identify the presence of MetS by correlating with the diagnostic, anthropometric and biochemical criteria proposed by the various societies for the diagnosis of MetS. Evaluating the indices is more practical in the daily than determining the levels of PAI-1, MCP-1 and Resistin which are not easy to access in public institutions (lines 366-371).

The title of the study is ambiguous, because once the body of the manuscript is read, it is observed that the population studied is not only obese children but also overweight and obese, in the same way a cardiometabolic risk assessment was not made per se, but only its association with the metabolic syndrome. The function for which the words “adipokines, lipoprotein ratios, and inflammatory markers” are placed in the title is not clear, because the reader understands that these markers are the ones associated with cardiometabolic risk, but in reality it was never carried out, only the lipoprotein indices were associated with metabolic syndrome, the title will have to be restructured, following the recommendations of STROBE, as well as indicating the study population, the dependent and independent variables and the type of study.

Once the STROBE guidelines were reviewed, it was decided to change the title of the document to: Metabolic, inflammatory and adipokine differences on overweight/obese children with and without metabolic syndrome: a cross-sectional study.

There are no references to the definitions presented, the references to each definition must be attached.

The reference used for the diagnosis of metabolic syndrome in children was added, this defines each of the components and cut-off criteria for metabolic syndrome (line 102)

Throughout the manuscript, the terms “abdominal adiposity” and “hypertension” are presented, which are mentioned in the results, but were never defined as such in the methodology, this error must be resolved.

The definitions for the diagnosis of metabolic syndrome according to the Ferranti criteria have been added in the material and methods section (lines 103 - 106).

The symbol ± is used throughout the text to show the standard deviation, please avoid it and place the standard deviation only in parentheses or with the abbreviation "SD".

The expression of the standard deviation has already been unified.

In line 202 of the text the word associated is mentioned to show the relationship between certain variables, I consider the term “correlated” to be more optimal.

We appreciate the comment and believe that the suggested term is more appropriate. The change has been made.

Reviewer 2

Please specify the type of study the authors used (E.g., Cross-sectional recruitment).

We appreciate the observation, and this has already been corrected in the material and methods section (line 78)

Please provide a more detailed context of the center of reference where the authors recruited the patients.

Information from the hospital where the study was conducted has been added (lines 81 – 84)

For this reviewer, it is quite unclear why the authors sought to explore a cut-off point to determine TGL/HDL and TC/HDL with and without MetS. Overall, I could interpret this to explore a tress-hold to define as a proxy of insulin resistance using MetS as their outcome variable in children. Please explain the purpose of this analysis. Furthermore, I would suggest to perform a stratification of this threshold among children and adolescence.

We appreciate the observation, but we consider that stratification by age group (children and adolescents) would not be appropriate because the population is not large enough to make this division.

The authors state that part of the inclusion criteria where patients ≥85th percentile of BMI (definition of overweight), but in table 1 they displayed 21 patients with normal weight (patients <85th percentile?). Please clarify.

In the material and methods section (lines 89-91) the groups studied were adequately described. Overweight and obese children were included and the results were contrasted with a group of normal weight children that was the control group

A main issue with table 1, is that the authors sought to compare the cardiometabolic risk using as a proxy the MetS construct, which sought to identify the profile among the subjects that fulfill the Ferranti criteria. A better approach to describe the results could be to display the overall population (n=107) comparing between subjects with MetS (n=64) and without MetS (n=43).

It has already been described that overweight children present morbidities more frequently than children with normal weight and healthy (Skinner AC, et al 2008). Generally, in studies where differences are established, such as those of the present study, the results are compared separately and are counteracted with a group healthy control to demonstrate the effect of overweight/obesity on the presence of MetS (Gökçe S, et al. 2013; Donma M, et al. 2015).

Skinner AC, Mayer ML, Flower K, Weinberger M. Health status and health care expenditures in a nationally representative sample: how do overweight and healthy-weight children compare? Pediatrics. 2008 Feb;121(2):e269-77. doi: 10.1542/peds.2007-0874. Epub 2008 Jan 14.)

Gökçe S, Atbinici Z, Aycan Z, Cınar HG, Zorlu P. The relationship between pediatric nonalcoholic fatty liver disease and cardiovascular risk factors and increased risk of atherosclerosis in obese children. Pediatr Cardiol. 2013 Feb;34(2):308-15. doi: 10.1007/s00246-012-0447-9. Epub 2012 Aug 9. PMID: 22875138.

Donma M, Karasu, E., Ozdilek, B. et al. CD4+, CD25+, FOXP3+ T Regulatory Cell Levels in Obese, Asthmatic, Asthmatic Obese, and Healthy Children. Inflammation 38, 1473–1478 (2015). https://doi.org/10.1007/s10753-015-0122-4

Another mayor issue is that the manuscript does not fully demonstrate the objective posed by the authors. As for what this reviewer understood, the sought to describe the metabolic and biochemical characteristics of patients with either obesity/overweigh with MetS and without MetS. Then, they sought to evaluate the adipokine and inflammatory profile, for which they found that patients with MetS had increased inflammation markers and lowest Adipo/Lep, which goes in line that MetS is mainly attributable to IR. Then, to prove this, the authors sought to evaluate the use of TGL/HDL and TC/HD-C ratios, for which found a cut-off value to determinate MetS. The structure of the results seems more like the objective was to describe metabolic, inflammatory and adipokine differences on overweight/obese children with and without MetS.

We understand the confusion and after discussing with the co-authors and reviewing the bibliography, we modified the objective of the work in the hope that it would be more understandable and descriptive, in addition to correlating with the results described. These changes are noted in the abstract (lines 3-5) and in the introduction (lines 74-75).

The aim of the study was changed to: Describe the metabolic, inflammatory and adipokine differences in overweight/obese children with and without metabolic syndrome.

I would suggest exploring a logistic regression model to assess whether metabolic, inflammatory and adipokine parameters predicts MetS.

Bivariate and multivariate logistic regression analyses were conducted to determine factors associated with MetS, variables with p<0.05 in bivariate analysis were included in multivariate analysis.

The only variables that showed a correlation were HDL and TGL. This information was added to section Material and Methods (lines 148-150) as well as in Results (lines 183-186) and discussion (lines 327-332).

Please describe the rationale and purpose of including the variables in Table 4

This analysis was used with the objective of showing that these indices are indeed useful to suspect or identify the presence of MetS by correlating with the diagnostic, anthropometric and biochemical criteria proposed by the various societies for the diagnosis of MetS. Evaluating the indices is more practical in the daily than determining the levels of PAI-1, MCP-1 and Resistin which are not easy to access in public institutions (lines 366-369).

A consideration would be to evaluate whether the causality of having MetS are all the metabolic dysfunctions caused by environmental or nutritional factors linked by the burden of obesity observed in Mexico.

A direct antecedent for this study is the results previously published by the national health survey where the main factors that contribute to obesity are described, these are the same ones that affect the population studied in our report.

Please let the conclusion statement at the end of the manuscript

A conclusion was added at the end of the work (lines 386 - 388)

Please round all the result for two decimals

The expression of the decimals has already been unified

The p-value in table 1 are not displayed in the pdf proof.

The margins of the table were modified for the visualization of the p value in table 1.

Please include headings in the methods and result section to separate the main ideas across the manuscript.

This has been done

Suggest including the waist-to-height ratio, waist-to-hip ratio, and non-HDL cholesterol estimation as part of table 1.

This study only evaluates the components of the metabolic syndrome, so we do not consider it necessary to include the other indices mentioned.

The estimated ratios in table 2 could be included as part of table 1. We think it is a good idea to put the data together. Tables 1 and 2 have been merged.

The ratios estimated in table 2 could be included as part of table 1.

We think it's a good idea to put the information together. Table 1 and 2 have been merged.

In figure 1, please modify hypertension to high-blood pressure, TGL to hypertriglyceridemia, fasting glucose to hyperglycemia, HDL to hypoalfaproteinemia. Also suggest including an additional label with the actual prevalence.

We appreciate the comments, the suggested changes were made including the term "hypoalphaproteinemia", although this is a more descriptive term and it is understood that in the criteria for metabolic syndrome the definition includes low HDL (HDL ≤ 40 mg/dl). We consider your suggestion and make the change.

In figure 3, please include the trending p-value to observe a potential dose-relationship progression with this two markers and Ferranti criteria.

The figure and figure caption have been modified.

Please check for typos of TC-HDL-C in table 3 and 4.

This expression has already been unified.

---

## [Decision Letter · Decision Letter 1]

17 Nov 2022

PONE-D-22-12285R1Metabolic, inflammatory and adipokine differences on overweight/obese children with and without metabolic syndrome: a cross-sectional studyPLOS ONE

Dear Dr. Cordero Pérez,

Thank you for submitting your manuscript to PLOS ONE. After careful consideration, we feel that it has merit but does not fully meet PLOS ONE’s publication criteria as it currently stands. Therefore, we invite you to submit a revised version of the manuscript that addresses the points raised during the review process.

We look forward to receiving your revised manuscript.

Kind regards,

Omar Yaxmehen Bello-Chavolla, MD, PhD

Academic Editor

PLOS ONE

Journal Requirements:

Additional Editor Comments (if provided):

Authors should just address the minor concerns put forth by Reviewer 2 prior to acceptance of the manuscript.

Reviewers' comments:

Reviewer's Responses to Questions

**Comments to the Author**

1. If the authors have adequately addressed your comments raised in a previous round of review and you feel that this manuscript is now acceptable for publication, you may indicate that here to bypass the “Comments to the Author” section, enter your conflict of interest statement in the “Confidential to Editor” section, and submit your "Accept" recommendation.

Reviewer #1: All comments have been addressed

Reviewer #2: All comments have been addressed

2. Is the manuscript technically sound, and do the data support the conclusions?

Reviewer #1: Yes

Reviewer #2: Yes

3. Has the statistical analysis been performed appropriately and rigorously? 

Reviewer #1: Yes

Reviewer #2: Yes

4. Have the authors made all data underlying the findings in their manuscript fully available?

Reviewer #1: Yes

Reviewer #2: No

5. Is the manuscript presented in an intelligible fashion and written in standard English?

Reviewer #1: Yes

Reviewer #2: Yes

6. Review Comments to the Author

Reviewer #1: My comments were answered correctly and the changes suggested in the manuscript were made. I have no additional comments

Reviewer #2: I would like to congratulate the efforts from the authors to improve the content and address the issues by both revisors in this manuscript. Significant and mayor improvements have been made. I have some specific issues that could be addresses within minor comments.

Results

• Please, move the p-value at the end of each sentence. Eg. “(p=0.002, OR=0.88, 95% CI=0.81-0.95)” should be “(OR=0.88, 95% CI=0.81-0.95, p=0.002)”

• In the Adipo/Lep ratio 6.68(7.89), what does the number in parenthesis represents? If it’s the standard deviations, please clarify.

• In table 3, please round to three decimals the correlation coefficient and the p-value. Additionally, please include the 95% confidence intervals within both types of correlations.

• In Page 19 and Line 349-350: The label “Fig 1. Frequency of the Ferranti criteria used for the diagnosis of metabolic syndrome.” seems out of line.

• In figure 1, please specify in the x-axis that this is the prevalence.

Although the conclusion passed by the authors is supported by their results, it does not go in line with the main objective. In a brief sentence, please give a response to the following question: what were the metabolic, inflammatory and adipokine differences among patients with overweight/obese children with and without MetS? After responding this question, the authors could include the statement in the original conclusion.

7. PLOS authors have the option to publish the peer review history of their article (what does this mean?). If published, this will include your full peer review and any attached files.

Reviewer #1: **Yes: **Ashuin Kammar García

Reviewer #2: **Yes: **Neftali Eduardo Antonio-Villa, MD

---

## [Author Response · Author response to Decision Letter 1]

30 Dec 2022

• Please, move the p-value at the end of each sentence. Eg. “(p=0.002, OR=0.88, 95% CI=0.81-0.95)” should be “(OR=0.88, 95% CI=0.81-0.95, p=0.002)”

We appreciate the observation, and this has already been corrected in the lineS 180 to 182

• In the Adipo/Lep ratio 6.68(7.89), what does the number in parenthesis represents? If it’s the standard deviations, please clarify.

We appreciate the observation, and this has been clarify in the Table 1. 

• In table 3, please round to three decimals the correlation coefficient and the p-value. Additionally, please include the 95% confidence intervals within both types of correlations.

The expression of the decimals has already been unified, and 95% confidence intervals have been added.

• In Page 19 and Line 349-350: The label “Fig 1. Frequency of the Ferranti criteria used for the diagnosis of metabolic syndrome.” seems out of line.

Figure captions were inserted immediately after the first paragraph in which the figure was cited, and these captions were aligned to the text.

• In figure 1, please specify in the x-axis that this is the prevalence.

We appreciate the observation, and this has already been corrected

---

## [Editor Report · Decision Letter 2]

23 Jan 2023

Metabolic, inflammatory and adipokine differences on overweight/obese children with and without metabolic syndrome: a cross-sectional study

PONE-D-22-12285R2

Dear Dr. Cordero Pérez,

We’re pleased to inform you that your manuscript has been judged scientifically suitable for publication and will be formally accepted for publication once it meets all outstanding technical requirements.

Kind regards,

Omar Yaxmehen Bello-Chavolla, MD, PhD

Academic Editor

PLOS ONE

Additional Editor Comments (optional):

All comments have been adequately addressed.
---

## [Editor Report · Acceptance letter]

26 Jan 2023

PONE-D-22-12285R2 

Metabolic, inflammatory and adipokine differences on overweight/obese children with and without metabolic syndrome: a cross-sectional study 

Dear Dr. Cordero-Pérez:

I'm pleased to inform you that your manuscript has been deemed suitable for publication in PLOS ONE. Congratulations! Your manuscript is now with our production department. 

Kind regards, 

on behalf of

Dr. Omar Yaxmehen Bello-Chavolla 

Academic Editor

PLOS ONE